# Exploring the Determinants of Food Security in the Areas of the Nam Theun2 Hydropower Project in Khammuan, Laos

**Phouvong Phami** [1], **Jianhua He** [1,2,*], **Dianfeng Liu** [1,2], **Su Ding** [1], **Patrik Silva** [1], **Chun Li** [3,4] **and Zhijiao Qin** [1]

1    School of Resource and Environmental Sciences, Wuhan University, 129 Luoyu Road, Wuhan 430079, China; phamiphouvong@yahoo.com (P.P.); liudianfeng@whu.edu.cn (D.L.); suding@whu.edu.cn (S.D.); patrisilva19@hotmail.com (P.S.); qinzhijiao@whu.edu.cn (Z.Q.)

2    Key Laboratory of Geographic Information System, Ministry of Education, Wuhan University, 129 Luoyu Road, Wuhan 430079, China

3    Asia International Rivers Center, Yunnan University, Guanxi Rd., Kunming 650091, China; lichun1020@whu.edu.cn

4    Yunnan Key Laboratory of International Rivers and Transboundary Eco-Security, Yunnan University, Guanxi Rd., Kunming 650091, China

*    Correspondence: hjianh@whu.edu.cn

**Abstract:** This article examines the driving forces of food security in the areas of the Nam Theun2 Hydropower Project (NT2) in Khamuan, Laos. A questionnaire survey was conducted to collect data from 100 NT2 resettlement households based on the random sampling technique. A linear regression technique was used to identify the influence of household food insecurity. The result showed that household size, food price, drought, shock, household income per month, number of laborers, gender of the household head, and farmland areas are important factors for household food insecurity. Policies should focus on irrigation that will permit yearlong cultivation. This will in turn become the stimulus for a concatenation of events in the process of development. People will resettle to practice agriculture while also expanding non-agricultural employment. Businesses in skills training, fish processing, textile, services, and crafts will be created, boosting household income. With inevitable population expansion, education in family planning will also be necessary to control population in relation to available resources.

**Keywords:** food security; linear regression; Laos; Nam Theun2 Hydro power

## 1. Introduction

Although much progress has been made to reduce the global undernourishment evidenced by a decline, between 1992 and 2016 from 23% to 13%, of the world's undernourished population, food security is among the greatest challenges humanity faces and continues to face in upcoming years [1]. From 2010 to 2012, about 870 million (12.5%) of the world's population were experiencing chronic hunger [2] especially in sub-Saharan Africa and southeast Asia [3], with most living in remote areas of developing countries which primarily depend on water and forest resources [4]. In parallel, these geographical regions are also expected to register the fastest population growth in the near future [5] which may compromise the desired scenarios. Laos is a country framed in this particular described scenario, which experiences both factors, with a population growth of 1.53% in 2019 [6] and with 23.2% of the population living below the national poverty line [7]. In addition, the climate change and its associated extreme weather conditions may result in crop and property destruction, which challenge

producers to shift to new regions, as has happened with the Nam Theun2 Hydropower Project in Khammuan, Laos. This engenders short-term fluctuations in food availability.

Food security is an indicator of food resource abundance, quality of life, and household well-being. It includes three aspects: food supply, food access, and food utilization [8]. It also stands for consistent and reliable access to sufficient food [9]. These three components can be measured by multiple indicators. However, there is not a set of predefined variables to measure food security, given the different impacts in different geographical contexts. For example, drought in a specific region may not affect their food security, if the people are not dependent on their own food production, but instead, are highly dependent on export from other regions. Food security is an international and national responsibility, which requires geographical regions to undertake meticulous research in arriving at their own understanding of this global issues. Therefore, understanding the driving factors that affect food security is crucial for decision-makers in order to tackle its adverse impact in this growing population.

There are some studies that have been done on food security in Laos [10–12], but they are either for the whole of Laos or for a specific province i.e., for Xekong and Savannakhet [4,13]. Besides the survey data, some studies also utilized secondary data to conduct studies on food security [14]. A determinant of household food security in resettled areas in Xekong was carried out [13]. The study applied logistic regression model to examine the factors influencing food security. The household size was one significant factor found that had a negative correlation with the household food security status. In addition, [4] conducted a study in Savannakhet and found that the factors influencing household food security were shock, gender of the head-of-household, and farmland area.

Several previous scientific publications related to household food security have also been investigated from various countries including Zimbabwe, Tanzania, Niger, Ethiopia, Pakistan, Kenya, Mexico, Thailand, Nigeria, Myanmar, Bangladesh, Indonesia, South Africa, and Sudan. For example, [15] used linear regression to examine the food security in Zimbabwe and he found household income as an important determinant of food security. A study by [16] in Mareko, Ethiopia concluded that farmland area was positively associated with household food security. Zakari et al. [2] conducted research in Southern Niger and the results have shown that the gender of the head-of-household and number of laborers in the household were significant and positively associated with household food security. On the other hand, drought variable was also one-factor influencing household food security in Tanzania [17]. The determinants of food security in rural Pakistan carried out by [18], showed that food price and household size significantly and negatively correlated with household food security.

From these studies varieties of variables were used, however the list of factors influencing household food security still needs to be extended bringing new contribution from novel geographical contexts. To the extent of our knowledge, currently, there is a lack of study related to food security in the region under consideration. This research, thus adds significance of this field of study.

This study provides the extent of knowledge, which might deepen our understanding of food security and its driving factors. It may be particularly useful to Laotian policymakers to improve their strategic decisions and to minimize the adverse impact of lack of food security. Based on the questionnaire and in loco observation data, we employed linear regression to identify the potential driving's factors of household food security, in the areas of the Nam Theun 2 Hydropower Project in Khamuan, Laos. The social-economic, biophysical-environment, and characteristic of the household was also taken into account.

*Food Security in Laos*

Laos is located in the Mekong river basin in Southeast Asia with abundant water and forest resources [4]. Shifting cultivation is one of the most important livelihoods in terms of food security [19]. According to the Lao Expenditure and Consumption Survey 2012/2013, about 23.2% of Lao people live below the national poverty line [7]. To develop the economy and to improve the quality of life, the government has combined households from various ethnic groups [20] (i.e., Khmu, Hmong, Phouthai, Tai, Makong, Katang, Lue, Aka and others) and scattered villages in the remote highlands to lowland

areas and along roads [13]. In 2008, the Laos government accepted the national nutrition policy, and as part of the directive, a framework has been developed to improve nutrition by 2020 [14]. This policy aimed to increase nutrition by protecting and facilitating the country's food production and supply by developing environmental and social regulations by implementing laws to protect household food security.

One of Laos' primary goals is to eliminate poverty. To do this, the government has been committed to promoting hydropower, and agricultural development [21]. In 2005, The Nam Theun2 (NT2) hydropower project was approved for construction by the World Bank and began full operation in 2010 after five years of construction. The project is located along the Nam Theun River, a tributary of the Mekong River in the central part of Laos. The main features of the project include the construction of a dam and the construction of a 450 km$^2$ reservoir in the Nakai Plateau, a catchment area of 4013 km$^2$. The installed capacity is 1070 MW, of which 93% is exported to Thailand and 7% is used for domestic consumption. The project is expected to generate approximately $1.9 billion in revenue for the government during the 25-years project concession period [22].

The World Bank's promotion of NT2 was considered a successful model for poverty alleviation and the construction of other large dams [23]. The project has assisted to improve living conditions through agriculture, fisheries, livestock, forestry, and non-agricultural activities [22]. Furthermore, the project provided road access to each village, good-quality domestic water, village offices, markets, primary schools, and residences for non-local teachers' and their relatives. The project also provided residential land and houses, electricity, farmland, and a specific compensation for the lost goods (e.g., rice and fruit trees) [24].

Most of the residents expressed satisfaction since the infrastructure and facilities are better when compared to those in their old villages [22]. Along with the benefit, however, the project has substituted the affected communities' livelihood, specifically the loss of traditional livelihoods, limited access to paddy farmland, natural resources, forests, and pastures. It was noted that the residents were not so satisfied with the land available for agriculture in the resettlement areas when compared to their old villages, the newly paddy farmlands had poor soil quality, and there was lack of or minimal access to irrigation. The re-settlers were unable to produce sufficient rice and food in order to keep their livestock; as pasture was not provided; most large animals died due to a lack of pasture and to flooding. For this reason, many families were no longer able to raise livestock, and many of them had decided to sell, and some stopped raising livestock. The other difficulties that re-settlers faced in restoring their livelihoods have forced them to switch from slash-and-burn farming to intensive agriculture with small plots of farmland is one of the factors restricting livelihood adaptation. Baird and Shoemaker who investigated experiences of people by visiting the NT2 resettlement areas in 2001 and 2014 found that their rice production had significantly decreased [23]. Off-farm agricultural training was conducted, but in most cases, the training was not fruitful, and given that the NT2 is a trans-basin hydroelectric in which electricity is generated through the transfer of water from the Nam Theun to Xebangfai River, there continues to be serious concerns associated with the downstream effects. The NT2 has greatly reduced the diversity and quantity of fishes and other aquatic organisms in Xebangfai and its tributaries. This is because various species of Xebangfai fishes migrates in and out of its tributaries [25].

## 2. Methodology and Data

### 2.1. The Study Area

The study was carried out in seven resettlement villages along the Nam Theun2 Hydropower Project in Khammuan Province, Laos. The area lies between longitude 17°20′ and 18°0′ and latitude between 105°0′ and 105°20′ (Figure 1). It has a total population of over 3720 inhabitants. There is a single-lane road access to every village. The total number of households in the study area is 966 of which 55.65% are women. The age group is mainly 15–65 years old, accounting for 62.72%. People engaged in fishing, hunting, and collecting of non-timber forest product (NTFP)s accounted for the

main economic activity, at 65%, while agriculture (on the received land) has come into the second place (15%). Other occupations are from non-farm owned businesses, occasional and light work, government official posts, and being permanently employed in non-agricultural activities. Cassava and root plants are the main crops cultivated in this area using traditional farming practices. The most commonly used agricultural tools include hand tools, which are available in almost all households. Food crops are grown for household consumption, and only the rest is sold in the local market. Animals like cattle, buffalos, pigs, and goats are raised and used directly by households. Most of the residents of the villages have primary level of education, followed by uneducated, secondary and post-secondary education with 54%, 22%, 20%, and 2% respectively.

**Figure 1.** Location of study areas (Nakia Nam Theun 2 Hydropower Reservoir).

The main food sources for household consumption come from agricultural production and market purchasing. The main agricultural products are fish (96%), bamboo and mushrooms (91%) vegetables (89%), fruits (80%), root plant (77%), and maize (72%). Some of the produce are sold to cover expenses but most of the household products are not enough for consumption needs over the season. The most important foods bought from the market are sugar, oil, milk, and butter (100%), meat (92%), glutinous rice (87%), white rice (70%), and eggs (59%).

*2.2. Conceptual Framework*

According to the FAO, there are three dimensions or pillars to food security, it includes food availability, food accessibility, and food utilization [26]. Food availability refers to the physical existence of food whether it is produced, purchased from the market or other sources [16]. Food accessibility is ensured when all individuals are able to obtain sufficient resources to acquire appropriate foods for a nutritious diet [27]. Food utilization refers to the intake and digestion of sufficient and quality food to maintain health, the proper use of food, the need for adequate energy and nutrients as well as food storage, processing, basic nutrition, and child-care knowledge and illness management [28]. To better understand how the above-mentioned food security is reflected in the study areas of the Nam Theun2

Hydropower Project, a conceptual framework presented in Figure 2 explains the relationship between food security status and indicator system.

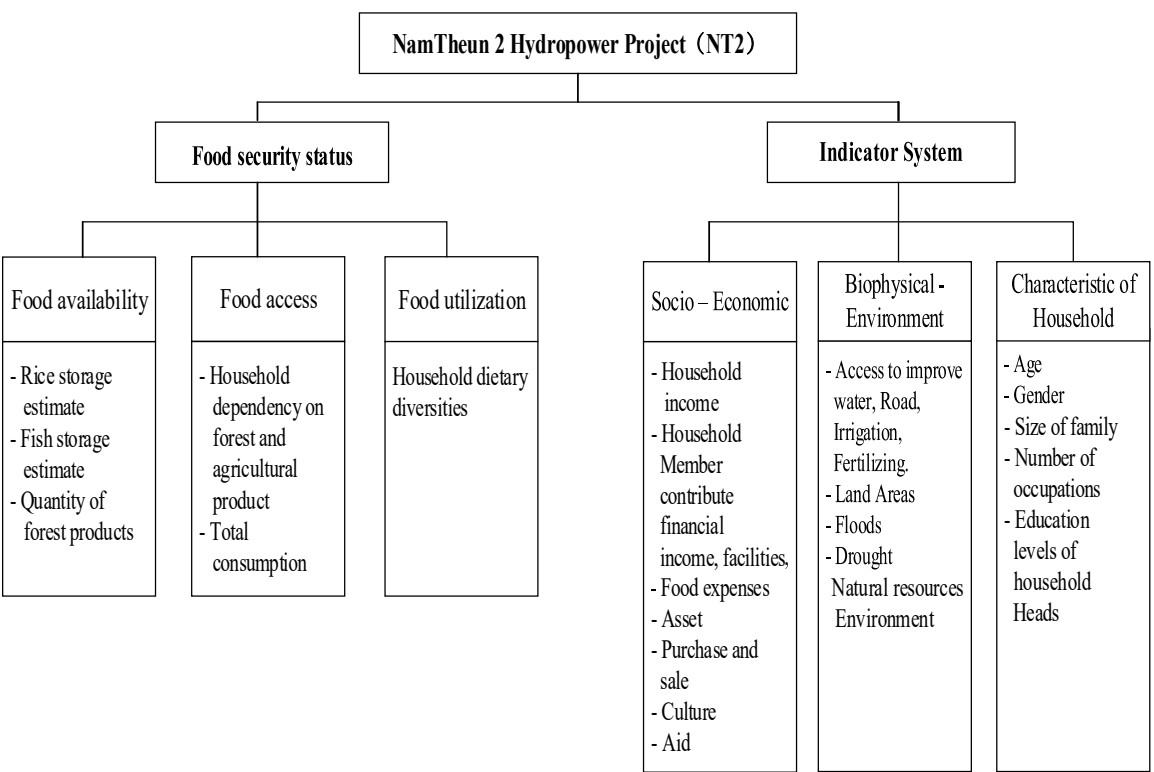

**Figure 2.** Conceptual Framework of Household food security.

### 2.2.1. Food Security Status

Food security status under this study was the food consumption score. This score was based on dietary diversity, food frequency, and the relative weight of food with nutritional importance consumed in a defined period by a household as shown in Table 1. For each household, the food consumption score is calculated by multiplying each food group frequency by each food group weight and then summing these scores into one composite score [29]. From the composite score we defined a threshold value shown in Table 2 to classify the household food security level.

**Table 1.** The calculation of food consumption score.

| No | Food Item | Food Groups | Days Eaten Last 7 Days (A) | Mass (B) | Score (A × B) |
|----|-----------|-------------|----------------------------|----------|---------------|
| 1 | Maize, rice, and other cereals | Staple | 7 | 2 | 14 |
| 2 | Cassava, potatoes, sweet potatoes, Peas, groundnuts, and cashew nuts | Pulses | 1 | 3 | 3 |
| 3 | Vegetables, leaves | Vegetable | 2 | 1 | 2 |
| 4 | Fruits | Fruit | 2 | 1 | 2 |
| 5 | Beef, poultry, pork, eggs, and fish | Meat and Fish | 2 | 4 | 8 |
| 6 | Sugar and honey | Sugar | 4 | 0.5 | 2 |
| 7 | Oils, fats, and butter | Oils | 2 | 0.5 | 1 |
| 8 | Milk, yogurt, and other diary | Milk | 0 | 0 | 4 |
| | Food Consumption Score | | | | 36 |

**Table 2.** Food security level.

| Food Consumption Score (FCS) | Food Security Level |
| --- | --- |
| 0 to 21 | Poor food consumption |
| 21.5 to 35 | Borderline food consumption |
| >35 | Acceptable food consumption |

### 2.2.2. Indicator System

The indicator system includes socioeconomic, biophysical-environments, and characteristics of the households. These factors appear to have an influence on household food security in the study areas. The following variables were selected to analyze whether they explain a household's food security or not.

It is possible that food security is constrained by large household sizes [15]. The increase in household size brings more pressure on consumption, than it contributes to the production [30]. Household labor reflects the resources available to a household, and thus a large labor force is more likely to ensure food security because they can execute agriculture activities timely. The gender of the household head is an important determinant of food security for the probability exists that households headed by a female is likely to have food insecurity. Livestock provides food high in micronutrients and protein, thus improving the dietary diversity and food consumption of households, while land-farming households are likely to produce cover crops that are easily sold for income and increases access to food better than households with no land. Income is a factor influencing household food security. Households which manage to secure a larger income is likely to have better access to the food they need and greater chances of being food secure [31]. Similar to the remittance as an alternative source of income, households with access to remittances can purchase more appropriate and nutritious foods and likely to be food secure than those without this source of income [32]. Better educated household-heads are more likely to receive information and use it in decision-making than those with less-learned heads, it is supposed that educated households are likely to benefit from agricultural technologies and to become food secure [15]. Additionally, household-heads' age is an important factor in decision making and it seems to directly impact on household food security [33]. It is expected that households that had experienced shock might accordingly experience inadequate dietary intake [34]. Religion and traditional knowledge influence food and nutrition security by shaping community eating habits, food preferences, household food distribution patterns, child feeding habits, food processing and preparation techniques, health, and hygiene practices [35]. Floods and drought can seriously damage lives, property, crops, infrastructure [36]. Increased drought make household less likely to have food security [17]. Food price is an additional factor that influences household food security. Rising food price decreases purchasing power and affects household food security [18]. Households located at a longer distance from farmland is another factor influencing household food security [37].

### 2.3. Data Collection

The seven villages investigated in this study were chosen based on our empirical evidence of dissimilarity and representatively among them. The inquired household was chosen by using a systematic sampling technique—counting houses along roads, and the household head was inquired if they were present. The questionnaire was pre-tested before the actual survey in order the readjust the questions and reduce the inconvenience during the field survey. Then, during two weeks a total of 100 households were surveyed by five inquirers. The questionnaire tried to encompass information on qualitative and quantitative data by collecting information about age, gender, education, family size, number of family laborers, and religion. Social and economic variables such as remittances, livestock ownership, farmland area, income, and sources of food were also collected. Furthermore, they were asked about the biophysical-environment variable that includes flooding, drought, environment change, and distance from home to farmland (see questionnaire in Supplemental Material).

## 2.4. Model Specifications

The linear regression model, known as ordinary least squares (OLS) regression, was fitted to a dataset with a dependent variable measured on a continuous scale and one or more independent variables [34]. The model is of the form:

$$Y = \beta_0 + \sum_{n=1}^{n} \beta_i X_i + \varepsilon_i \tag{1}$$

where $\beta_0$ is a constant term, $\beta_i$ is the regression coefficients, $X_i$ is the explanatory variable. The explanatory variables used in this study model were: age, gender, education level of household head, household size, number of household labor, religion, remittances, livestock ownership, farmland area, food price, household income, shocks (big problem affecting the household including causes of illness, death of household member, house damage, unusually heavy rainfall, member left the household, accident, crop pests, disease, price increases, etc.,), flood, drought, and distance from home to farmland. (See Table 3) $\varepsilon_i$ is the error term. Y is food security status (dependent variable); food security status is using food consumption score (FCS); the formula of the FCS calculation:

$$FCS = \sum_{i=1}^{n} X_i, \ n = 1 \sim 8 \tag{2}$$

$$X_i = W_i F_i \tag{3}$$

where *i* represents one food item, a total of eight groups of food. *W* represents the weight of food and *F* is the frequency consumption (seven days' worth), the categorized of *FCS* is a continuous data. The variable in the model taking the value of 1 if the household is poor in food consumption; 2 if the household is borderline in food consumption, and 3 if the household is acceptable in food consumption.

**Table 3.** Explanatory variables and their description. HH: household.

| Variables | Description and Measurement |
| --- | --- |
| Age ($x_1$) | Age of HH head in a number of years |
| Gender ($x_2$) | D = 1 if HH head is male; 0 = otherwise |
| Education ($x_3$) | D = 1if HH head is literate; 0 =otherwise |
| Household size ($x_4$) | Number of household members |
| Number of labor ($x_5$) | Number of Labor in household |
| Religion ($x_6$) | D = 1 if HH Buddhist; 0 = otherwise |
| Remittances ($x_7$) | Households have remittances in cash |
| Livestock ownership ($x_8$) | Number of HH livestock |
| Farm Land ($x_9$) | Actual land size in hectares |
| Food price ($x_{10}$) (Lao KIP) | D = 1 if HH food insecurity is caused by food price; 0 = otherwise |
| Income ($x_{11}$) (Lao KIP) | Total household income |
| Shocks ($x_{12}$) | D = 1 if HH food insecurity caused by shock; 0= otherwise |
| Flood ($x_{18}$) | D = 1 if food insecurity is caused by flooding; 0 = otherwise |
| Drought ($x_{19}$) | D = 1 if food insecurity is caused by drought; 0 = otherwise |
| Distance from home to farmland ($x_{21}$) | Actual distance in Km |

## 3. Results and Discussion

### 3.1. The Relationship between Food security and Its Driving Forces

The linear regression results show the relationship between household food security and several independent variables presented in Table 4. The model explained about 70% of the variation on food security status, as expressed by adjusted R square.

**Table 4.** The outcomes of the linear regression model (dependent variable: food security status).

| Variables | Coefficient | Variables | Coefficient |
|---|---|---|---|
| (Constant) | 1.861 *** | Farmland | 0.077 ** |
| Household size | −0.088 *** | Food price | −0.153 ** |
| Number of labor | 0.077 ** | Income per month | $1.062 \times 10^{-7}$ *** |
| Gender | 0.188 ** | Remittances | $−5.968 \times 10^{-9}$ |
| Age | 0.000 | Flooding | −0.034 |
| Religion | 0.006 | Drought | −0.260 *** |
| Education | −0.094 | Shock | −0.164 ** |
| Livestock | −0.001 | Distance from home to farmland | 0.178 |
| R | 0.865 | | |
| R Square | 0.748 | | |
| Adjusted R square | 0.704 | | |
| Std. Error of the Estimate | 0.265 | | |
| Sig. F Change | 0.000 | | |

Note: The sign of the coefficients indicate the type of relationship between independent variables and food security status. Positive sign indicates that an increase in the independent variable is associated with an increase in food security status and a negative sigh suggests that and increase in the independent variable is associated with a decrease in food security status. The F-statistic value and its associated *p*-value show the statistical significance of the models. ***. Significant at the 0.01 level (2-tailed). **. Significant at the 0.05 level (2-tailed).

From our model we found that eight variables has a statistically significant effect on the household food security status. Specifically, household size, food price, drought, and shock have shown a negative relationship with food security status. On the other hand, household monthly income, number of laborers in the household, gender of household head, and farmland areas have shown a positive relationship with household food security status. Variables such as age, religion, education of household head, livestock ownership, remittances per year, flooding, and distance from home to farmland were not statistically significant in our model.

The results show that as the household size increases, the probability of food security decreases. In other words, the large sized of the household is more likely to be food insecure than in small sized household. This was expected because an increase in household members means more people eating from the same resources; hence the household members may not be able to get enough food to fully satisfy themselves [38].

The food security status of households tends to rise with an increase in the number of households' labor. This result was expected, since an increasing number of labors in a household may increase the amount of resources in that particular household and positively impact their food security. In addition, household labor is needed for farming activities and the labor in the household is crucial to the family's work on the farm, especially for the processing of land preparation, weeding and harvesting to expand food productivity and fishing, hunting, collecting, or logging [2].

Male-headed households are more food secure compared to female-headed households. This result is supported by [39] findings. Moreover, in our study area, we also found that female-headed households are relatively poorer with most of them being widows or older. Another reason might be the fact that historically, activities like farming, fishing, and hunting are normally done by men, which make female-headed households less productive in these activities.

Household food security increases as farm size expands. As farm size increases, farmers take more interest in agriculture and are likely to further improve their activities and products which tends to improve food security status, as also concluded by [40] in his study conducted in Benue State, Nigeria.

Drought has been one of the most important constraints to crop production, which has had an impact on food security worldwide. Results have shown that households which have experienced droughts in the last eight years tend to present a decrease in food security. Drought is associated with a high production loss or even neglect of farmlands, thus affecting household food availability. Although, our result supported by [17] findings, shows that drought tends to decrease the food security, on the

other hand, it contrasts with [9] findings. This is because in his study area (rural northern hinterland of Pakistan) most of the households were net purchaser of food (86%), which makes them less sensitive of drought year.

As found by [4] our results show that the probability of households being food secure decreases with increasing shock. This association was expected because of shock/illness of a household member, a member leaving the household, accidents, crop pests, and disease, which all have the affect of price increase-results in the expenditures of the households to address the shocks, such as for medicine or hospital bills, and decreased yields.

The greater the income of the household, the higher the probability was of a household being food secure. This was expected because an increase in income means greater access to food, which is also supported by [8] showing that household income was statistically significant and had a negative relationship with the household food security status.

Food price is also one of the factors influencing household food security. Similar to [18] our results have shown that an increase in the food price is associated with decreases in household food security. According to the dataset, the percentage of the households that spent their income on food, including food ingredients (i.e., oil, fat, milk, sugar, salt, butter, etc.) was 100%, meat (92%), glutinous rice (87%), white rice (70%), and other food item such as eggs, corn, root plants, fruit, vegetables, mushrooms, and fish (59%, 27%, 22%, 20%, 10%, 8% and 4%, respectively). Therefore, any increase in food prices will affect the purchasing power of households and may also affect their food security status and well-being.

On the other hand, seven variables—age, religion, education of the household head, livestock ownership, remittances per year, flooding, and distance from home to farmland were not statistically significant in our experimental tests as shown in Table 4.

The fact that age of the household head was not statistically significant in our model may be related to the homogeneity in most of the observed household, with more or less similar household composition, which may result in little average age variability. The religion in the study area was predominantly Buddhism (63%), with the remaining, mostly, only believing in God, but with no specific religion. We believe that this fact does not significantly affect food security, due to religious homogeneity in the study area. Similarly, the level of education of the household head was not statistically significantly related to the household food security status. Although, this result is contradictory to earlier expectations, that non-educated household head may be disadvantaged in terms of access to food, our empirical observation and surveyed data revealed that most of the household head were the elderly (70%) who had low education level (i.e., primary school or less) and the number of household head with university degree was almost insignificant (2%). This fact results in a little variation in household head education level in our study area, which may not have a great impact on household food security. In addition, the household head may be non-educated or have a low level of education, but younger members of the household may have secondary school education, which may also cause an imbalance our results. Likewise, livestock ownership was not statistically significant in our model. There were 66% of the households who owned livestock (cattle, buffaloes, pigs, and goats) for draft power, selling, or direct consumption. However, because we merged livestock that may provide source of income and those that are kept by the household by historical and cultural reasons, this may also unbalance the effect of livestock relationship with household food security.

Remittances from migrated relatives and friends, seem to be important, especially for the households with insufficient labor or limited dietary diversity and practiced eating fewer meals regularly to relief hunger. However, it was not statistically significant in our model. Only nine in one hundred households in our study area benefit from remittances. In addition, we observed that families who do receive remittances apparently presented better living conditions, which may be an indication of higher income. In this case, income that was already included in our regression model may be a substitute for or take into account remittances. Therefore, if the model accounts for income, then remittances alone may be irrelevant in explaining variations in household food security status. The distance from home to farmland was not statistically significant in our model due to the fact that the

farmlands are relatively close to the household home with maximum distance around approximately 2 km. This may result in no difference in working time in farmland which may not greatly influence productivity and income or ultimately food security status.

This study has its limitations. The reduced sample size considering the number of independent variables used in the model might have constrained the number of independent variables which are statistically significant in the model. In addition, the fact that all the categorical variables were recoded into dummy variables (two categories) might have affected their significance in the model, since this approach may not have captured the heterogeneity present in such variables. Therefore, we encourage that similar studies with larger sample sizes should be conducted in order to improve the quality of the modelled relationship.

### 3.2. Policy Implications

According to the finding of this study, some suggestions for improving the food security of the households in the resettlement villages were proposed.

1. As the drought factor was associated with loss of food production, especially rice, maize, pulses, and vegetable, the policy should focus on irrigation so that households can carry on with cultivation throughout the year and increase their food availability.
2. The rapidly growing population should be controlled through family planning and education. This could be utilized to reduce the number of household members, consequently decreasing consumption imbalances and ultimately the effect on household food security.
3. Families should also be encouraged with training and provided with off-farm businesses such as skill training, fish processing, textiles, service business and other handicrafts which were identified as potential sources of nonagricultural income that can be generate income.
4. It is imperative to encourage re-settlers to raise livestock, and to grow fruits and vegetables in order to increase food consumption and protein intake. This also necessary to expand non-agricultural employment opportunities for villagers. Health education should additionally be promoted in an effort to reduce shock and household expenses.
5. The farmland provided for every household is important and so efforts should be made to avoid sale of such necessary lands. At the same time, there is a need to provide and promote the use of fertilizers especially the organic fertilizer. There is also the need to promote strategies such as crop diversity, and low-cost supply of inputs like fertilizer and improved crop varieties with full management practices.
6. Female-headed households are more vulnerable to food insecurity than male-headed households. Thus, the policy should be focused on social allowances and assistance targeting such families during food scarcity.
7. Sufficient labor resources will have positive impacts on household food security. To increase productivity, households need to be particularly encouraged in the training of new agricultural techniques and labor skills.

The main effects of the NamTheun Hydro-power Project in Nakai region are enhanced in Figures 3–5, with a great consequence in food security status.

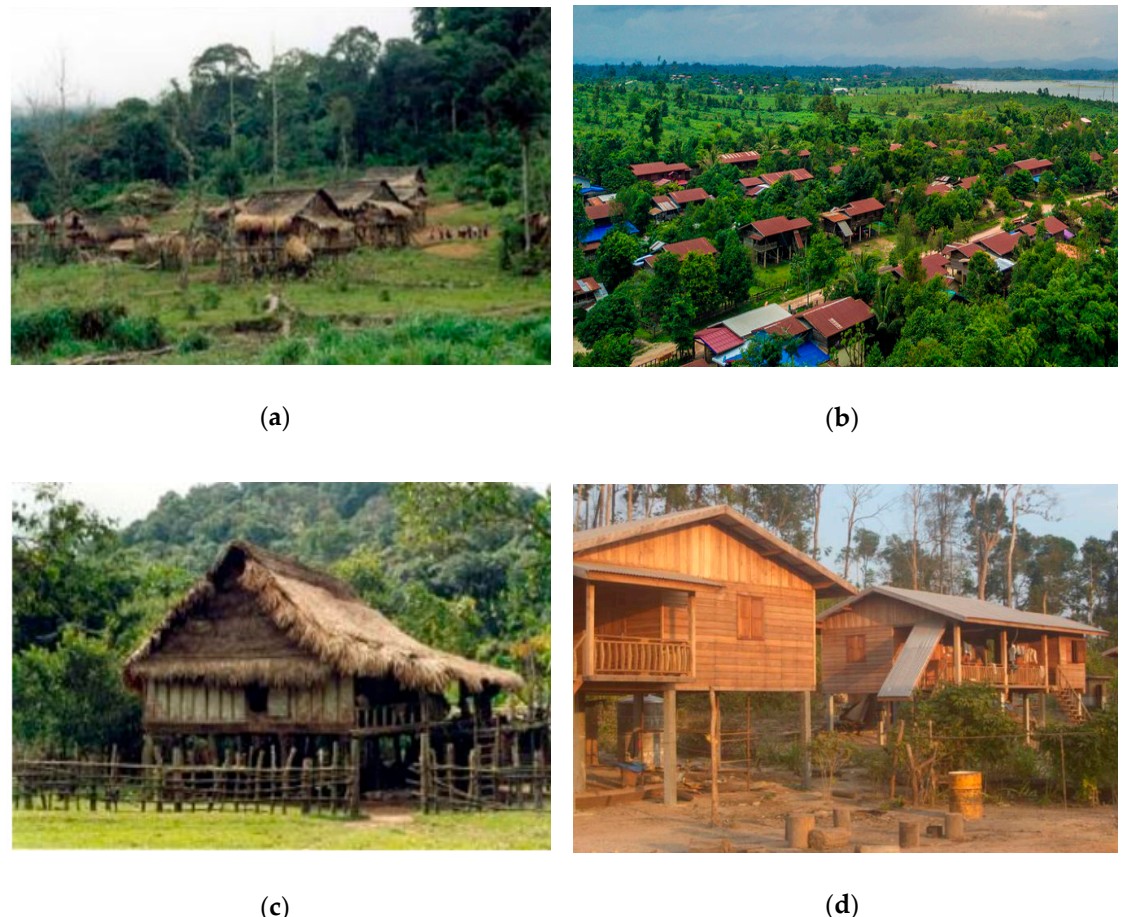

**Figure 3.** The effect of the NamTheun Hydro-power Project: (**a**) Old village before the NamTheun Hydro-power Project; (**b**) new resettlement village after the NamTheun Hydro-power Project; (**c**) housing before NamTheun Hydro-power Project; and (**d**) housing after NamTheun Hydro-power Project.

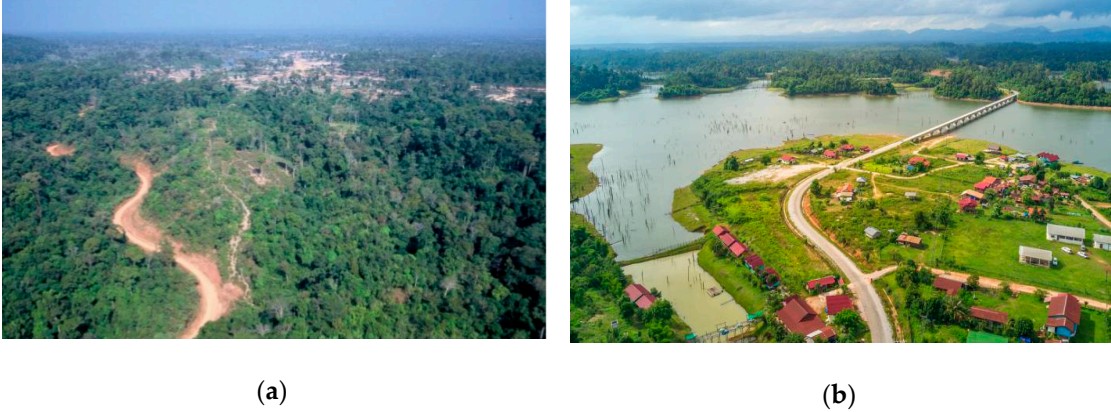

**Figure 4.** The effect of the NamTheun Hydro-power Project: (**a**) the main access roads before the NamTheun Hydro-power Project; and (**b**) the main access roads after the NamTheun Hydro-power Project.

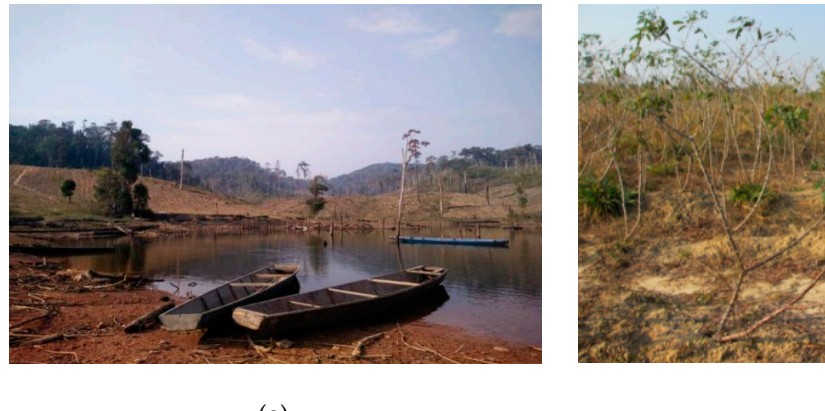

(**a**)                                                                    (**b**)

**Figure 5.** The effect of the NamTheun2 Hydro-power Project: (**a**) local port for fishing; and (**b**) soil degradation for planting close to resettlement villages after the project.

## 4. Conclusions

The purpose of this research was to examine the determinants of food security in the areas of the Nam Theun2 Hydropower Project in Laos. Based in our methodological experiment and analysis, we found that the number of laborers in the household, the gender of the household head, the farmland area, and the household monthly income have a positive relationship with household food security status. In other words, an increase in these variables is associated with an increase in household food security status. On the other hand, variables such as household size, drought, food price, and shock are negatively associated with household food security status, meaning that an increase in these variables is associated with a decrease in household food security status. Furthermore, these factors not only influence household food security but also negatively affect their livelihood.

This study provides foundations and guidelines for policymakers, especially for Laos, for effective interventions in food security in rural communities, particularly in shifted communities after a natural disaster or any other shock, in order to have quick resilience and to minimize the adverse impacts.

**Supplementary Materials:** The following are available online at http://www.mdpi.com/2071-1050/12/2/520/s1, Questionnaire for Household and Village Head.

**Author Contributions:** Conceptualization, P.P., J.H. and D.L.; data curation: P.P.; formal analysis: P.P., S.D., P.S., and Z.Q.; investigation: P.P.; methodology: P.P., J.H., D.L., P.S., S.D., and Z.Q.; project administration: P.P.; resources: J.H.; software: S.D., P.S., and C.L.; supervision, J.H., D.L., C.L., and P.S.; validation: D.L. and P.S.; writing—original draft, P.P.; writing—review editing, D.L., P.S., and S.D. All authors have read and agreed to the published version of the manuscript.

**Funding:** This work was financially supported by the National Natural Science Foundation of China (grant number: 41871301 and 41771429) and the National Key R&D Program of China (grant number: 2018YFB0505402).

**Acknowledgments:** We cordially thank Nakai district office and the head of villages and communities for a warm welcome and for providing information which enabled to undertake this study, and for the Savannakhet University, Laos, for providing students which helped us during the field data collection. We thank Junlong Huang and colleagues of the Lab 40–School of Resources and Environment, Wuhan University, China, for the guidance offered during the initial stage of this research. We also thank Henok Girmatsion, Tab Gatdiet Bannak Theng and Kymmala Francis for the English editing in the final manuscript.

**Conflicts of Interest:** The authors declare no conflict of interest.

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
