# Peer review of "Exploring the Determinants of Food Security in the Areas of the Nam Theun2 Hydropower Project in Khammuan, Laos"

_sustainability, doi:10.3390/su12020520_

Round 1
Reviewer 1 Report
The paper analyzes the determinants of food security in an area of Laos where a Hydropower project was implemented. The paper needs substantial improvements before being published.
Introduction and literature review are, at the moment, condensed into a single paragraph. Both should be distinguished and developed better. In particular, the link that the authors intend to explore between the project mentioned in the title and the theme of food security should be clarified.
Paragraph 2.2.1. introduces a food consumption score but does not contextualize it within the conceptual framework specified in the previous paragraph. The title of paragraph 3.2 does not correspond to the content. The authors try, unfortunately not succeeding, to explain the model and the relationship between this and the Hydropower project. Lines 283-290 offer a very superficial analysis.
Author Response
Dear Chief of Editors,
Thank you very much for your valuable comments and suggestions. We have revised the manuscript accordingly, and please we would like to re-submit it for your consideration and appreciation. We have addressed the comments raised by the reviewers, and the amendments are highlighted in red in the revised manuscript. Point by point responses to the reviewers’ comments is listed below in this letter.
This manuscript has been edited and proofread by all authors and substantial English improvement have been made.
We hope that the revised version of the manuscript is now acceptable for publication in your journal.
We look forward to hearing from you soon.
Yours sincerely,
Phami
Responses to Reviewer #1
Reviewer 1
## Comments and suggestion 1
The paper analyses the determinants of food security in an area of Laos where a Hydropower project was implemented. The paper needs substantial improvements before being published.
## Response 1:
We have made a substantial improvement in the whole paper. We have added new sentences and clarify many points there were not soo clear.
## Comments and suggestion 2
(1) Introduction and literature review are, at the moment, condensed into a single paragraph. (2) Both should be distinguished and developed better. (3) In particular, the link that the authors intend to explore between the project mentioned in the title and the theme of food security should be clarified.
## Response 2:
(1) We have created many paragraphs from the existing paragraphs making the text less condensed and much clear, as it can be seen it the revised manuscript. (2) In addition, we have reformulated the whole introduction section, making it more logical and concise. The introduction and the literature review can be now easily distinguished. However, we think is more appropriated to let them stay in the same section, without any further subsection.
(3) More precisely, the link between the Nam Theun2 Hydropower Project and food security was clarified, as suggested.
## Comments and suggestion 3
(1) Paragraph 2.2.1. Introduces a food consumption score but does not contextualize it within the conceptual framework specified in the previous paragraph. (2) The title of paragraph 3.2 does not correspond to the content. (3) The authors try, unfortunately not succeeding, to explain the model and the relationship between this and the Hydropower project. (4) Lines 283-290 offer a very superficial analysis.
## Response 3:
(1) Please, we would like to kindly suggest you review the mentioned paragraph. The food consumption score is contextualized within the conceptual framework. The variables mentioned within the boxes (food availability, food access and food utilization) were used for the calculation in food consumption score. The conceptual framework must be read together with the table 1 for a better understanding.
(2) Thank you! We agreed with your suggestions. We have removed the title of subsection 3.2 since it is the continuation of subsection 3.1 “The relationship between food security and its driving forces”. Therefore subsection 3.2 was not necessary.
(3) We would like to kindly remember you that the main objective of this paper was to explore the determinant of food security in the areas of Nam Theun2 Hydropower Project. Therefore, it not the main objective to explore the relationship between the project and food security. However, we have made significant changes in the interpretation, explanation and analysis of the model results, showing the relationship between each factor (independent variables) and food security status (dependent variable).
(4) We strongly agree with you that the analysis from line 283 to 290 was very superficial. Therefore, major revisions from [Lines 379-421] were made to deepen the interpretation and discussion of the results, as can be seen in the body of the manuscript.

Reviewer 2 Report
Overall, the topic is interesting as it explores the relationship between human activities in a hydropower project and the food security of communities in Laos. The authors will need a professional review of the English grammar, I have attempted to highlight in yellow on the manuscript where changes are necessary.
The title may be revised as "Exploring the determinants of food security in the areas of the Nam Theun 2 Hydropower Project in Khamuan, Laos" instead of "Exploring the determinants of the food security in the areas of the Nam Theun 2 Hydropower Project in Khamuan, Laos".
L41: which ethnic groups?
L43-46: The sentence can be revised with some reference to food security
L48: to be written in full as: Nam Theun2 (NT2) hydropower project.
L50: construction instead of constructed
L60-61: Please revise the grammar for clarity
L78-79: Revise for clarity
L81: There are some studies that have been done on food security in Laos [15-17], but they are either for the whole of Laos....
L84: ...in Xekong was carried out [8], the study applied a logistic regression model to examine the factors influencing food security. Household size...
L88: Several ......related to household food security have also been investigated from various countries...
L91: ..used linear regression to examine the food security in Zimbabwe.
L93: ....'associated'
L95: positively associated
L98: by [23], they showed that price and household size were significant and negatively correlated.....
L101-106: Please revise for clarity
L109: seven resettlement villages along the Nam Theum2 hydropower project?
L113: full stop before "People engaged in fishing, hunting, collecting non-timber food products (NTFPs), which accounted......
Does collecting, rearing and harvesting of insects part of this NTFPs?
L121: directly by the households. Revise the next sentence (L121-123) for clarity.
L137: According to the FAO, there are four dimensions or pillars to food security, it includes stability.
L186: Please revise the "Data collection" for clarity.
L202: please change the symbol as above for consistency.
L208: as above for the error term
L210: FCS instead of FSC
L213-214: 'poor in food consumption; 'borderline in food consumption'; 'acceptable in food consumption'
L220: approximately
L225: capital letter 'H'.
L235-238: Please revise for clarity.
L242-249: labour or labor? Need to be consistent.
L255-256: Kindly state the facts from the cited reference [38] and relate it to your findings.
L259-262: Revise for clarity. What are the differences from their findings and suggest why this is so?
L273-276: Kindly revise for clarity.."According to the dataset, most households spent their income....."
L294: Remittances from where?
L299: From our field-work observation, most of the agricultural areas and their residents...
L302: According to the findings of this study,....
L304: 1. As the drought factor was associated with loss of food production...
L305-306: ....can carry on with cultivation..
L311: 3. Families should also be encouraged with training and provided with off-farm businesses...
L327: 7. A sufficient labor resources will have positive impacts on household..
L330-337: A short sentence to summarize Figures 3, 4 and 5.
L339-344: Revise for clarity. "The important findings which influence household food security..."

Author Response
Dear Editor-in-Chief,
Thank you very much for your valuable comments and suggestions. We have revised the manuscript accordingly, and please we would like to re-submit it for your consideration and appreciation. We have addressed the comments raised by the reviewers, and the amendments are highlighted in red in the revised manuscript. Point by point responses to the reviewers’ comments is listed below in this letter.
This manuscript has been edited and proofread by all authors and substantial English improvement have been made.
We hope that the revised version of the manuscript is now acceptable for publication in your journal.
We look forward to hearing from you soon.
Yours sincerely,
Phami
Responses to Reviewer #2
Overall, the topic is interesting as it explores the relationship between human activities in a hydropower project and the food security of communities in Laos. The authors will need a professional review of the English grammar, I have attempted to highlight in yellow on the manuscript where changes are necessary.
The title may be revised as "Exploring the determinants of food security in the areas of the Nam Theun 2 Hydropower Project in Khamuan, Laos" instead of "Exploring the determinants of the food security in the areas of the Nam Theun 2 Hydropower Project in Khamuan, Laos".
We highly appreciate your kind attention. As the title has been revised above we agree with your opinion. The title is "Exploring the determinants of food security in the areas of the Nam Theun 2 Hydropower Project in Khamuan, Laos" instead of "Exploring the determinants of the food security in the areas of the Nam Theun 2 Hydropower Project in Khamuan, Laos".
L41: which ethnic groups?
[Line 95 page 3] We have added: “various ethnic groups such as lao, khmu, hmong, phouthai, tai, makong katang, lue, aka and other”.
L43-46: The sentence can be revised with some reference to food security
We agree with your opinions and the sentence has been revised with reference to food security and strategy to protect household food security. As we have reorganised the paragraph [Line 43-46] became to [Line 97-101 Page 3]. New sentence was “ In 2008, the Laos government accepted the national nutrition policy, and as part of the directive, a framework has been developed to improve the nutrition by 2020[13]. This policy aimed to increase nutrition by protecting and facilitating the country's food production and supply by developing environmental and social regulations by implementing laws to protect household food security.”
L48: to be written in full as Nam Theun2 (NT2) hydropower project.
As we have reorganized the paragraph [Line 107 page 3] Nam Theun2 (NT2) hydropower project has been revised instead of “NT2”
L50: construction instead of constructed
[Line 109 Page 3] construction has been revised instead of “constructed”
L60-61: Please revise the grammar for clarity
Thanks for your suggestion. In the revised version, we expressed it in the form of a statement: [Line 121-123 Page 4] The project also provided residential land and house, electricity; farmland, and a specific compensation for the lost goods (e.g., rice and fruit trees).
L78-79: Revise for clarity
In the revised version, we expressed it in the form of a statement: [Line 140-142 Page 4] “The NT2 has greatly reduced the diversity and quantity of fishes and other aquatic organisms in Xebangfai and its tributaries”.
L81: There are some studies that have been done on food security in Laos [15-17], but they are either for the whole of Laos (...)
We are agreeing with you. As we reorganised the paragraph, [Line 56-57 Page 2] “There are some studies that have been done on food security in Laos[2-4], but they are either for the whole of Laos” has been revised instead of “There are some studies have been done on food security in Laos [15-17], but they are either for whole Laos”
L84: ...in Xekong was carried out [8], the study applied a logistic regression model to examine the factors influencing food security. Household size...
[Line 58-60 Page 2] “... in Xekong was carried out[5], the study applied logistic regression model to examine the factors influencing food security. The household size” have been revised instead of “....in Xekong carried out by[8], the study was applied a logistic regression model to examine the factors influencing food security, household size...”
L88: Several ...related to household food security has also been investigated from various countries...
We have been reordered the paragraph [Line 64-65 Page 2] ” Several previous scientific publications related to household food security have also been investigated from various countries including Zimbabwe” has been revised instead of “Several previous scientific publications related to this study have also been investigated from various countries including Zimbabwe”
L91: ..used linear regression to examine the food security in Zimbabwe.
[Line 67 Page 2] “used linear regression to examine the food security in Zimbabwe” has been revised instead of “used linear regression examined the food security in Zimbabwe”
L93: ...' associated'
[Line 69 Page 2] “associated” has been revised instead of “ associated”
L95: positively associated
[Line 71 Page 2] a term of “positively associated” has been revised instead of “ positive associated”
L98: by [23], they showed that price and household size were significant and negatively correlated.....
[L74 Page 2] a phrase of “.... by[6], they showed that food price and household size were significant and negative correlated with household....” has been revised instead of”...by[23], food price and household size were significant and negative correlated ...”
L101-106: Please revise for clarity
[Line76 Page 2-Line 87 Page3] “Therefore, from these studies varieties of variables were used, however the list of factors influencing household food security still need to be extended bringing new contribution from new geographical contexts. In addition, to the extent of our knowledge, currently, there is a lack of study related to food security in the region under consideration in this study, which add significance of this study..
This study provides knowledge extent which might deepen our understanding of food security and its driving factors, particularly useful to Laos policymakers to improve their strategic decisions and minimize the adverse impact of lack of food security. Based on the questionnaire and in loco observation data, we employed linear regression to identify the potential driving’s factors of household food security, in the areas of the Nam Theun 2 Hydropower Project in Khamuan, Laos. The social-economic, biophysical-environment and characteristic of the household was also taken into account.” has been revised instead of “ [Line 101-106]”.
L109: seven resettlement villages along the Nam Theum2 hydropower project?
Yes, there are total seventeen resettlement villages along the Nam Theum2 hydropower project but we have selected just seven villages to be a target of this study.
L113: full stop before "People engaged in fishing, hunting, collecting non-timber food products (NTFPs), which accounted......
[Line 177 Page 5] full stop before "People engaged in fishing, hunting, collecting non-timber food products (NTFPs), which accounted....” has been revised instead of “comma”
Does collecting, rearing and harvesting of insects part of this NTFPs?
Yes, we have count the insects as one of NTFPs part.
L121: directly by the households. Revise the next sentence (L121-123) for clarity.
[Line 185 Page 5] “directly by the households” has been revised instead of “directly by households”
[Line 185-188 Page 5] “Most of the residents of the villages have primary level of education, followed by none educated, secondary and after secondary with 54%, 22%, 20% and 2% respectively.” has also been revised and instead of” Education of the villager is mainly holding a primary level, followed by none educated, secondary and after secondary exert at 54%, 22%, 20% and 2% perspective. ”
L137: According to the FAO, there are four dimensions or pillars to food security, it includes stability.
Thank you for your kind suggest and comment , this point we agreed with you . In this paper we have cited from “Stamoulis, K. and A. Zezza, A conceptual framework for national agricultural, rural development, and food security strategies and policies. 2003”
[Line 198 Page 6] We have revised intead the” Food security has three dimensions“ by “According to the FAO, there are three dimensions or pillars”
L186: Please revise the "Data collection" for clarity.
Data collection has been carefully revised [Line 262-277 Page 9] was the revised version.
L202: please change the symbol as above for consistency.
[Line 282 Page 9] the symbol of “β0” in the equation has been revised instead of “β0”
L208: as above for the error term
[Line 289 Page 9] has been corrected
L210: FCS instead of FSC
[Line 290 Page 9]” FCS” revised instead of “FSC”
L213-214: 'poor in food consumption; 'borderline in food consumption'; 'acceptable in food consumption'
[Line 293-295 Page 10] 'poor in food consumption; 'borderline in food consumption'; 'acceptable in food consumption' has revised instead of “ poor food consumption”;” borderline food consumption”;”acceptable food consumption”
L220: approximately
[Line 301 Page 10] “approximately” has been revised instead of “approximate”
L225: capital letter 'H'.
[Line 306 Page 11] capital letter 'H' has been revised instead of “h”
L235-238: Please revise for clarity.
[Line 316-318 Page 11] “The results shows that as the household size gets larger, the probability of food security decreases. In other words, the large size of the household is more likely to be food insecure than in small size household.” has been revised instead of “The variable household size gets larger, the probability of food security decreases. In other words, the large size of the household is likely to be food insecure than small size household”
L242-249: labour or labor? Need to be consistent.
[Line 325 Page 12] “labour” we decided to use”labor”
L255-256: Kindly state the facts from the cited reference [38] and relate it to your findings.
As we have revised and proofed the manuscript, some references need to be cited, so cited number 38 become to number 40. [Line 341-342 Page 12] “as also concluded by [40] in his study conducted in Benue State, Nigeria ” has been revised instead the state:“ This result is related to [38] ”
L259-262: Revise for clarity. What are the differences from their findings and suggest why this is so?
[Line 343-353 Page 12] We have revised and re-explained as” Drought has been one of the most important constraints to crop production, which has had an impact on food security worldwide. Results have shown that household which has experimented drought in the last eight years tends present a decreasing in food security. Drought is associated with a high production loss or even leaving the farm uncultivated, thus affecting household food availability. Although, our result is supported by[17] findings, showing that drought tends to decrease the food security, on the other hand, it contrasts with[8] findings. This is because in his study area (rural northern hinterland of Pakistan) most of the households were net purchaser of food (86%), which make them less sensitive of drought year. ”
L273-276: Kindly revise for clarity.."According to the data-set, most households spent their income....."
[Line 362- 371 Page 13] “Food price is also one of the factors influencing household food security. Similar to [18] our results have shown that an increase in the food price is associated with decreases in household food security. According to the dataset, most of the households spent their income on food, including food ingredients (oil, fat, milk, sugar, salt, butter, etc.) by 100%, meat (92%), glutinous rice (87%), white rice (70%), and other food item such as eggs, corn, root plants, fruit, vegetables, mushrooms, and fish (59%, 27%, 22%, 20%, 10%, 8% and 4%) respectively. Therefore, any increase in food prices will affect the purchasing power of households and may also affect their food security status and well-being” has revised instead of “Line 273-276”
L294: Remittances from where?
Response: [Line 405 Page 14] Remittances are from their migrated relatives and friends
L299: From our field-work observation, most of the agricultural areas and their residents...
Thank you very much for this suggestions [Line 405- 417 Page 14] We have revised and re – explanation. The phrase of “From our field-work observation, most of the agricultural areas and their residents...” has been removed.
L302: According to the findings of this study,...
[Line 423 Page 14] “According to the findings of this study“.... has been revised instead of ” According to the finding of the study...”
L304: 1. As the drought factor was associated with loss of food production...
[Line 425 Page 14] “1. As the drought factor was associated with loss of food production...” has revised instead of ”1. As a drought factor was associated with the loss of productions”
L305-306: ....can carry on with cultivation..
[Line 426-427 Page 14] ”.... can carry on with cultivation..”has revised instead of “... can carry on cultivation”
L311: 3. Families should also be encouraged with training and provided with off-farm businesses...
[Line 432-433 Page 14]: “3. Families should also be encouraged with training and provided with off-farm businesses...” has revised instead of” Families should also be encouraged in training and providing in off-farm businesses”
L327: 7. A sufficient labor resources will have positive impacts on household...
[Line 448 Page 15] ” Sufficient labor resources will have positive impacts on household..” has revised instead of“A sufficient labor resources will be positively impacted on household...”
L330-337: A short sentence to summarize Figures 3, 4 and 5.
[Line 452-455 Page 15] “Figure 3. The effect of NamTheun Hydro-power Project: (a) Old village before NamTheun Hydro-power Project; (b) New resettlement village after NamTheun Hydro-power Project; (c) Housing before NamTheun Hydro-power Project and (d) Housing after NamTheun Hydro-power Project.” has revised instead of “ Figure 3. Village and housing in Nakai Plateau: (a) Old village before NamTheun Hydro-power Project; (b) New resettlement village after NamTheun Hydro-power Project; (c) Housing before NamTheun Hydro-power Project; (d) Housing after NamTheun Hydro-power Project.”
[Line 456-458 Page 16] “Figure 4. The effect of NamTheun Hydro-power Project: (e) The main access roads before the NamTheun Hydro-power Project and (f) the main access roads after NamTheun Hydro-power Project.” has revised instead “Figure 4. (e) The main access roads in Nakai Plateau before NamTheun Hydro-power Project and (f) The main access roads in Nakai Plateau after NamTheun Hydro-power Project”
[Line 459 Page 16] “ Figure 5. The effect of NamTheun2 Hydro-power Project: (g) Local port for fishing and (h) Soil degradation for planting close to resettlement villages after the project” has revised instead “Figure 5. (g) Local port for fishing and (h) Soil degradation for planting close to resettlement villages after the project.”
L339-344: Revise for clarity. "The important findings which influence household food security..."
[Line 462-474 Page 16] a conclusion “The purpose of this research was to examine the determinants of food security in the areas of the Nam Theun2 Hydropower Project in Laos. The important findings which influence household food security status were that the number of labor in the household, the gender of household head, farmland area and household income per month were positively associated with food security, whereas household size, drought, food price and shock had a negative relationship. These factors not only influence household food security but also negatively affect their livelihood.” has been carefully revised.

Round 2
Reviewer 1 Report
thank you for addressing all the issues raised in previous review. I think the paper is publishable
Author Response
Dear Editors,
Thank you very much for your letter and advice. We have revised the manuscript, and would like to re-submit it for your consideration. We have addressed the comments raised by the reviewers. Point by point responses to the reviewers’ comments are listed below this letter.
This manuscript has been edited and proofread by an English speaker.
We appreciate very much your taking time in handling and consideration and we hope that the revised version of the manuscript is now acceptable for publication in your journal.
We look forward to hearing from you soon.
Yours sincerely,
Phouvong Phami, Jianhua He, Dianfeng Liu, Su Ding, Patrik Silva,Chun Li, Zhijiao Qin
24 December 2019
Responses to Reviewer #2
The title is now more appropriate and some revision to the English grammar was carried out. The methodology is adequate enough. However, the major concern with the paper is the level of the English language and its style. This is common in many parts of the paper. The language will need to be adequately revised by a native English speaker or professional to have a chance of being published.
Please follow the guidelines for the journal - see the style of references. Any limitation in the current study should be mentioned in addition to the suggestions.
Response 1: Thanks for your suggestion. The revised version was proofread by a native speaker with a professional background to avoid grammatical errors.
Response 2: The style of references has been carefully revised.
Response 3: We highly appreciate your kind attention and we strong agree with your opinion. We have mentioned the limitation [Line 444-450 Page 15].
For the results and conclusions, we have carefully read and tried to explain and clarify this part, we are based on the results of the model and our understanding to complete. If you find that there is need to improvement, we are happy to make a revise, and it would be nice if you could provide us with specific Suggestions.
Reviewer 2 Report
The authors have attempted to improve the paper "Exploring the determinants of food security in the areas of the Nam Theun 2 Hydropower Project in Khamuan, Laos".
The title is now more appropriate and some revision to the English grammar was carried out. The methodology is adequate enough. However, the major concern with the paper is the level of the English language and its style. This is common in many parts of the paper. The language will need to be adequately revised by a native English speaker or professional to have a chance of being published.
Please follow the guidelines for the journal - see the style of references. Any limitation in the current study should be mentioned in addition to the suggestions.
Author Response

(The authors gave the same response as above.)
